# A new method to retrieve relative humidity profiles from a synergy of Raman lidar, microwave radiometer and satellite

Chengli Ji[1], Qiankai Jin[2], Feilong Li[2], Yuyang Liu[2], Zhicheng Wang[1], Jiajia Mao[1], Xiaoyu Ren[1], Yan Xiang[4], Wanlin Jian[5,6] , Zhenyi Chen*[2,3] and Peitao Zhao*[1]

[1] CMA Meteorological Observation Centre, Beijing, 100081, China

[2] State Environmental Protection Key Laboratory of Food Chain Pollution Control, Beijing Technology and Business University, Beijing, 100048, China

[3] Key Lab. of Environmental Optics & Technology, Anhui Institute of Optics and Fine Mechanics, Chinese Academy of Sciences, Hefei, 230031, China

[4] Institutes of Physical Science and Information Technology, Anhui University, Hefei, 230031, China

[5] Sichuan Meteorological Observation and Data Center, Chengdu, 610072, China

[6] Sichuan Meteorological Observatory Heavy rain and Drought – Floor Disaster in Plateau and Basin Key Laboratory of Sichuan Province,Chengdu, 610072, China

*Corresponding author: E – mail: zychen@btbu.edu.cn (Zhenyi Chen),
peitaozhao@163.com (Peitao Zhao)

**Abstract** Precise continuous measurements of relative humidity (RH) vertical profiles in the troposphere have emerged as a considerable scientific issue. In recent years, a combination of diverse ground-based remote sensing devices has effectively facilitated RH vertical profiling, leading to enhancements in spatial resolution and, in certain instances, measurement accuracy. This work introduces a newly developed approach for obtaining continuous RH profiles by integrating data from a Raman lidar, a microwave radiometer, and satellite sources. RH profiles obtained using synergistic approaches are subsequently compared with radiosonde data throughout a five-month observational study in China. Our suggested method for RH profiling demonstrates optimal concordance with the best correction coefficients R of 0.94 in Huhehaote

(HHHT), 0.92 in Yibin (YB) and 0.93 in Qingyuan (QY), respectively. Accordingly, the mean bias (MB) reached the lowest values of 4.93% in HHHT, 2.63% in YB and 2.40% in QY. The mean value of RH decreased with height and presented seasonal characteristics in QY. Finally, the RH height-time evolution in a convective case was analyzed. This study firstly integrates satellite data into ground-based measurement to provide information on RH profiles in China, which may aid in further evaluating their regional characteristic and their impacts on the local ecosystem.

*Keywords:* relative humidity profiles, Raman lidar, microwave radiometer, satellite

## 1. Introduction

Relative humidity (RH) is a crucial parameter in characterizing aerosol-cloud interactions (Fan et al., 2007) and is necessary as input for weather forecasting models (Petters and Kreidenweis, 2007; Wex et al., 2008; Mochida, 2014). The combination of these RH profiles with aerosol optical data allows us to obtain hygroscopic growth factors for different aerosol types (Zieger et al., 2013; Granados et al., 2015). However, the temporal resolution of routine observations performed by weather services is rather low, typically with one or two radiosonde launches per day (Schmetz et al., 2021). And significant mesoscale weather phenomena, including the movement of frontal systems and the formation of convective boundary hygroscopic growth or clouds, transpire rapidly, making it more challenging to adequately monitor the evolution of atmospheric profiles (Kang et al., 2019; Long et al., 2023; Chen et al., 2024). Consequently, precise information with great temporal resolution is essential for examining these events.

The current Raman lidar technology enables concurrent measurements of temperature and water vapor mixing ratio profiles to derive RH profiles (Reichardt et al., 2012; Brocard et al., 2013). But it requires calibration by the use of collocated and simultaneous observations from a radiosonde or microwave radiometer (MWR) (Mattis et al., 2002; Madonna et al., 2011; Foth et al., 2015). In addition, the average error of Raman lidar is relatively small within the effective height range but limited in the higher height detection.

MWR is another way to provide atmospheric RH observations with high temporal resolution (Hogg et al., 1983; Ware et al., 2003; Zhang et al., 2024). Although MWR has a certain penetration ability for harsh weather conditions such as clouds, their vertical resolution and accuracy are not high, especially for RH which vary greatly (Xu et al., 2015). Thus it is challenging to deliver continuous high-resolution RH information with a single instrument. The synergy of complementary information from both active and passive instruments can provide a more comprehensive understanding of atmospheric processes (Stankov, 1995;

Furumoto et al., 2003; Delanoë and Hogan, 2008; Blumberg et al., 2015; Tuner et al., 2021). For example, when both Raman lidar and MWR are measuring collocated and simultaneously, continuous temperature, water vapor profiles and thus RH profiles can be obtained operationally (Navas-Guzmán et al., 2014; Barrera-Verdejo et al., 2016; Foth et al., 2017; Toporov et al., 2020). However, most of their algorithms primarily utilize statistical methods, performing data fusion between different instruments based on long-term time-series data from individual locations. While these approaches are suitable for observations at single stations, they lack universality when applied to scenarios requiring data integration from multiple sites or broader geographical coverage. Moreover, replacing instruments or equipment may also introduce additional inconsistencies.

For accurate RH profile retrieval at higher heights, satellites have global detection capabilities and are highly effective for oceanic skies and remote land areas (Zhang et al., 2022; Wang et al., 2023).For example, Wang et al. measured the subgrid-scale variability of critical relative humidity ($RH_c$) to investigate the cloud parameterization based on the diagnostics from CloudSat/CALIPSO satellite data. Some deal with the retrieval of the atmospheric layer averaged relative humidity profiles using data from the Microwave Humidity Sounder (MHS) onboard the MetOp satellite (Gangwar et al., 2014). Geostationary Operational Environmental Satellite (GOES)-13 and the Moderate-Resolution Imaging Spectroradiometer (MODIS) data are also be combined to estimate hourly relative humidity at the surface level (Ramírez-Beltrán et al., 2019). Another sounder, SAPHIR, onboard MEGHA-TROPIQUES provides measurements in six water vapour channels for sounding the atmospheric humidity. Brogniez et al. (2013) and Gohil et al. (2013) have shown the potential of SAPHIR sounder in retrieving the atmospheric humidity profile.

But the time resolution of polar orbit satellites is determined by the repeated coverage time of the satellite orbit (Skou, et al., 2022). A single satellite can generally only achieve repeated observations twice a day, and the time resolution is also relatively low. Furthmore, few observations are available from China's satellite Fenyun (FY), to the use of synthetic retrieval of RH information. This study aims to introduce a novel technique that integrates Raman lidar, MWR, and satellite data (FY4B) using an optimum estimating methodology. It is given with a focus on two aspects: i) Evaluation of the proposed synergetic method, and ii), investigation of the RH characteristics at different heights and in different geographic regions. This paper is thus structured as follows. Descriptions of the individual equipment is presented in Section 2. Section 3 illustrates the process of the new synergetic algorithm combining the ground-based

and satellite data. Section 4 presents the RH statistic results and its time-height evolution in a strong convective case. Finally, conclusions are summarized in Section 5.

## 2. Instrumentation

### 2.1 Raman lidar

The Raman lidar method can assess the water vapor mixing ratio profiles through inelastic backscatterring signals from nitrogen at 387 nm and from water vapor at 407 nm (Whiteman, 1992; Mattis et al., 2002; Adam et al., 2010). At the lowest height, the intersection of the laser beam with the receiver's field of view in the bistatic system is incomplete. Nevertheless, the overlap of both Raman channels is presumed to be equivalent; thus, the overlap effect could be minimal concerning water vapor measurements. The collected water vapor measurements, then along with concurrent temperature profiles from a co-located MWR allow us to obtain RH profiles. The vertical and temporal resolution of Raman lidar and other instruments are listed in Table 1.

### 2.2 Microwave Radiometer (MWR)

The Microwave Radiometer (MWR) serves as a passive instrument designed to measure atmospheric emissions across two frequency bands within the microwave spectrum (Cimini et al., 2006; Crewell and Löhnert, 2007). There are seven channels set along the 22.235 GHz $H_2O$ absorption line. Humidity information can be extracted from these observations. The seven channels of the alternative band from 51 to 58 GHz within the $O_2$ absorption complex encompass the vertical temperature profile data. Consequently, the fully automatic MWR enables the derivation of temperature and humidity profiles with a temporal resolution of up to 5 minutes. The method for inverting temperature and humidity profiles is the neural network method in this study. It uses statistical methods to optimize the long-term average radiosonde data and relies on previous radiosonde data (Yang et al., 2023).

### 2.3 Radiosonde data

We use radiosonde data from the China Meteorological Administration (CMA) station for reference analysis. It is located in the same place as the Raman lidar, and provides on-site measurements of atmospheric pressure, temperature, and RH. During the observing campaign, radiosondes were launched twice a day (08:00 LST and 20:00 LST). The height of the radiosonde balloon can be determined by the ascent time of the radiosonde balloon. The vertical resolution of the raw data is 3 m/layer. To match other data, the vertical resolution of the raw data is interpolated to 30 m (0-3000 m) and 250 m (3000-10000 m), respectively.

### 2.3 Satellite

In 2016 and 2021, China successfully deployed two second-generation geostationary meteorological satellites, Fengyun-4A (FY4A) and Fengyun-4B (FY4B), both equipped with the Geostationary Interferometric Infrared Sounder (GIIRS). The GIIRS therefore became the first geostationary orbiting meteorological satellite (Yang et al., 2023). This approach could achieve the detection of weather systems across China and its neighboring regions with high temporal and spatial resolution. So it enables a more comprehensive understanding of the atmospheric vertical structure, including the retrieval of atmospheric temperature profiles for 1000 m layers and moisture profiles for 2000 m layers (Yang et al., 2017), respectively. In comparison to FY4A, the GIIRS on FY4B exhibits a broader spectral range, improved spectral resolution in the long-wave IR band, and superior radiometric calibration accuracy and detection sensitivity (Sufeng et al., 2022). Specifically, the temporal resolution of GIIRS has enhanced from 2.5 hours for FY4A to 2 hours for FY4B, and the spatial resolution has progressed from 16000 m to 12000 m at nadir. The atmospheric humidity profiles utilized in this study, derived from GIIRS, are generated through the neural network algorithm created by the National Satellite Meteorological Centre (NSMC) (Bai et al., 2022). The data is available online: http://fy4.nsmc.org.cn/nsmc/en/ theme/FY4B.html (accessed on 12 December 2024).

## 3. Methods and evaluation

### 3.1 Lidar, MWR and satellite synergetic algorithm

This study aims to obtain a continuous time series of RH profiles by integrating ground-based remote sensing techniques, including Raman lidar, MWR, and satellite data, in a straightforward manner to facilitate a wide range of applications. The retrieval process involves a systematic four-step algorithm that integrates the Raman lidar water mixing ratio profile and MWR brightness temperatures along with satellite data. The retrieval framework is shown as in Fig. 1 and the retrieval process is detailed in the following paragraphs.

Step 1: Data quality control. Data with quality control codes of 0 and 1 for FY4B and 0 for ground-based remote sensing data is selected. The lidar only retains data with a SNR value greater than 3. The threshold value of the SNR is set as 3 based on our extensive comparisons with radiosonde data from CMA's long-term observations. The results indicate that selecting lidar signals with signal-to-noise ratios (SNR) >3 can significantly improve the consistency between retrieved RH profiles and radiosonde measurements. So in the data selection period, the Raman signal starts with the first SNR greater than 3 and ends with five consecutive SNRs less than 3. The real-time observing data are designated as $R_{radio}$, $R_{lidar}$, $R_{MWR}$ and $R_{satellite}$ in Fig. 2.

Step 2: Data spatial-temporal matching. This process aims to match the above quality-controlled data with the radiosonde data at a height of 0-10000 m in time and space before the synergetic algorithm. For the time matching, temperature from MWR and water vapor data from Raman lidar are selected corresponding to the radiosonde data time (00:80 LST and 20:00 LST). In terms of spatial matching, the FY4B data is selected from the nearest grid point to the ground observing station for the horizontal scale. The data at vertical heights are interpolated to the resolution of 30 m (0-3000 m) and 250 m (3000-10000 m).

Step 3: Correction coefficient determination. The deviation between the temperature and humidity data of satellites and ground-based remote sensing data at each height is quantitatively calculated and analyzed to prepare for the optimal stitching process in the next step. Here the deviation of lidar, MWR and FY4B is designated as $D_{lidar}$, $D_{MWR}$ and $D_{satellite}$, respectively.

$$D_{lidar}=R_{lidar}-R_{radio} \tag{1}$$

$$D_{MWR}=R_{MWR}-R_{radio} \tag{2}$$

$$D_{satellite}=R_{satellite}-R_{radio} \tag{3}$$

The correction coefficients $C_{lidar}$, $C_{MWR}$ and $C_{satellite}$ are calculated as follows

$$C_{Lidar}=(|D_{satellite}|+|D_{MWR}|)/[2*(|D_{satellite}|+|D_{MWR}|+|D_{lidar}|)] \tag{4}$$

$$C_{MWR}=(|D_{satellite}|+|D_{lidar}|)/[2*(|D_{satellite}|+|(D_{MWR}|+|D_{lidar}|)] \tag{5}$$

$$C_{satellite}=(D_{MWR}+D_{MWR})/[2*-(|D_{satellite}|+|D_{MWR}|+|D_{lidar}|)] \tag{6}$$

Step 4: Synergetic algorithm iteration and evaluation: Based on the above spatial-temporal data matching and correction coefficients calculation at different heights, a dynamic optimal stitching algorithm (Fig. 2) is conducted. To ensure the independence between the tested sample and the true value, the temperature and humidity profiles of the current time are fused using the correction coefficient of the previous time, and then compared with the radiosonde data at the same time for evaluation. The correlation coefficient (R), the root mean square error (RMSE), and mean bias (MB) are used as inspection indexes. Finally, the retrieved RH information $S_{RH}$ could be obtained through the following formula.

$$S_{RH}=R_{satellite}*C_{satellite}+ R_{MWR}*C_{MWR}+R_{lidar}*C_{Lidar} \tag{7}$$

From the process we can see that compared to these existing techniques, our new method not only incorporates satellite data but also dynamically determines optimal fusion coefficients. Because the fusion coefficients are dynamically determined by comparing the deviations from

other measurements with the reference of radiosonde, it highlights that this new algorithm is real-time calibrated. And it can guarantee the device model independence and geographical adaptability. Thus it eliminates constraints imposed by equipment specifications or observation locations, ensuring broad applicability across diverse scenarios.

*3.2 Error analysis*

To evaluate the performance of the synergetic algorithm for RH profiles, a comparative analysis was conducted between retrieved values and actual radiosonde measurements. Let N represent the total number of samples. The measured value is designated as Oi, with i representing the sample label. The value obtained through the new synergetic algorithm is designated as Gi. The evaluation indicators consist of MB, mean absolute bias (MAB) and RMSE are defined by the following formulas:

$$MB = \frac{\sum_{i=1}^{N}(G_i - O_i)}{N} \tag{8}$$

$$MAB = \frac{\sum_{i=1}^{N}|G_i - O_i|}{N} \tag{9}$$

$$RMSE = \sqrt{\frac{\sum_{i=1}^{N}(G_i - O_i)^2}{N}} \tag{10}$$

**4. Results**

*4.1 General statistic information*

A five-month data set has been chosen for a statistical analysis of RH profiles. The observation period spans from July 1 to November 30, 2024. The observing elements are RH data from 47 stations in China (yellow circles in Fig. 3) at the height of 0-10000 m. To investigate RH retrieval accuracy, we provide the comparison results of four methods (lidar, MWR, satellite, and synergetic algorithm) utilizing the radiosonde data as the reference at 47 sites in Table 2. Then Huhehaote (HHHT, northern China), Yibin (YB, middle China) and Qingyuan (QY, southern China) are selected as 3 representative sites (red stars in Fig. 3) for more detailed analysis, as shown in Fig. 4 and Table 3.

Generally, the synergetic algorithm at 47 sites presents the maximum correlation coefficient R value of 0.98 with the minimum RMSE of 5.27% in Table 2. For three representative sites, the regression line from the synergetic algorithm at all heights similarly provides the best fitting results, with the largest correlation coefficients R of 0.94, 0.92 and 0.93 in HHHT, YB and QY respectively (Table 3). The correlation coefficient R for lidar measurement follows with

marginally higher values of 0.83 in HHHT, 0.86 in YB and 0.86 in QY, indicating its greater applicability compared to other single instruments. MWR presents the lowest R of 0.74 and 0.80 in HHHT and YB, while performing better (R = 0.75) than that from satellite (R = 0.66) in QY. In terms of RMSE, the lidar-, MWR- and satellite-derived RH all show values larger than 18% at three sites. The synergistic use of a multi-source algorithm decreases the RMSE down to the lowest value of 10% in HHHT.

The regression line for lidar and MWR in HHHT, as illustrated in Fig. 4, exhibits a slope that is less than that of the one-to-one line. This implies that greater variations arise with increased RH in HHHT. Though the synergetic algorithm also presents similar trends, its RMSE decreased to 26% in HHHT. The regression line of MWR and lidar in YB and QY are larger than the one-to-one line, indicating the larger bias for less humid.

As RH vertical profiles are height-dependent, Fig. 5 presents the MB profiles observed at different heights in terms of four methods. Generally, the MB in the RH of lidar in the lower troposphere (below 3000 m) outperforms the other two single methods (MWR and satellite) at three sites. No significant biases between radiosonde and lidar are noticeable. Specifically, the lowest MB values (4.93% in HHHT, 2.63% in YB and 2.40% in QY) in the comprehensive region of the tropospheric region are achieved when lidar data is incorporated into the synergetic algorithm. This is because lidar is an active remote sensing technology with more accuracy compared to MWR and satellite. The lidar data's efficacy is enhanced at heights below 3000 m when integrated with data from other sources within the boundary layer.

However, the MB from lidar increased drastically above this height, up to the highest value 28.67% in HHHT, 29.91% in YB and 20.09 % in QY. It is reasonable that the atmosphere changes so fast that radiosonde do not assess exactly the same air mass as lidar. In the meantime, lidar is increasingly constrained at elevated heights because of a decreased SNR. Hence lidar is more trustworthy in the lower layer, i.e. below 3000 m.

In contrast, the MB from satellite (FY4B) over 3000 m varied steadily within the range of approximately 15% at three sites. Therefore the satellite data in the far height range would be more reliable and could be employed in the synergetic algorithm at higher layers. Compared to lidar and satellite, the MB from MWR gives the largest uncertainty in HHHT at all heights. This may result from the discrepancy between the temperature recorded by the radiosonde and that obtained from the MWR in HHHT. However, it yields relatively less variation than lidar and satellite in YB and QY. Anyway, the synergetic method gives the best result for over three observing sites at almost all heights. And accurate measurements of RH vertical profiles provided here are highly beneficial for analyzing the hygroscopic growth of local aerosols.

The sources of the discrepancy can stem from several aspects. First, although all instruments

are co-located in the ground, radiosondes deviate at higher heights, and signals can be

disrupted if clouds are present. Second, satellites provide gridded data, requiring the selection

of ground observation points closest to its grid's latitude and longitude, which introduces

uncertainty. Finally, both MWR and satellite are passive remote sensing technologies, which

are inherently less precise than active remote sensing. Besides the inherent hardware difference,

the errors during the retrieval process (e.g., neural networks for MWR) are also unavoidable.

*4.2 Mean monthly analysis*

RH mean monthly vertical profiles have been derived from the synergistic method illustrated

in Fig. 6. Because RH profiles were retrieved from water ratio profiles and temperature profiles.

For this property, the RH seasonal behavior may be more complicated. For example, no

obvious seasonal behavior of RH profiles is found in HHHT or YB. However, QY still

presents the most likely seasonal characteristic at most of the heights, with the highest mean

values in summer at 1000-2000 m (80.65% in July) and lowest values at 7000-10000 m in late

autumn (20.50% in November) in Fig. 6e-f. The elevated RH observed in QY's summer may

be related to the sufficient water vapor and large transport volume as QY is located in coastal

areas. So the characteristic of QY would be more dependent on water vapor.

For comparison, HHHT and YB are relatively random. Over 3000 m in HHHT (Fig. 6a-b), RH

in August shows predominantly high values with the highest value of 65.37% at 5000-7000 m.

Different from HHHT and QY, the RH profiles in November of YB interestingly show the

highest values (83.95%) in the lower atmosphere (0-1000 m) in Fig. 6c-d. It suggests the

reduced temperatures observed in autumn of YB promote proximity to saturation conditions,

resulting in elevated RH values in November. It is also worth noting that RH above 3000 m in

November of YB decreases dramatically as height increases, with the minimum RH of 13.91%

at 7000-10000 m. That could be explained by more rapid fluctuations in the water vapor

density and temperature in YB in the higher layer under the control of the subtropical monsoon

climate zone. Anyway, this plot illustrates a clear decrease in the RH values with heights at

three sites.

Though there is no obvious RH uncertainty caused by regional differences, we found that QY

exhibits the predominant seasonal feature throughout most heights. In contrast, no discernible

seasonal characteristics in RH profiles are observed in HHHT or YB. Thus we believe diverse

atmospheric circulation patterns and geographical environments could result in regional

variations in RH values.

*4.3 Case analysis*

We selected two different severe convective events in YB (one hailfall and one heavy precipitation) for comparison in Fig. 7. At 23:00 LST on April 15, a thunderstorm with strong winds and hail occurred. The synergetic algorithm retrieved RH profile showed that before 22:00 LST, the RH was high (around 90%) at 3000 m height, low (20%-50%) between 3500 m and 8000 m, and above 80% between 8000 m and 9000 m (Fig. 7a). This indicates that before the severe convection, the upper and lower layers were relatively moist, while the middle layer (3500 m-8000 m) was relatively dry (red arrow in Fig. 7a). Such a condition favors the evaporation and cooling of ice particles descending from the upper atmosphere, leading to refreezing and hail formation.

In contrast, the RH profile from 25 May to 26 May showed that the entire troposphere (0-10000 m) presented high RH values (>70%) starting at 19:00 LST, which was conducive to heavy precipitation (Fig. 7b). According to ground station observations, YB recorded an hourly rainfall of 52 mm at 21:00 LST, along with gale-force winds of 23 m/s (9th grade). Most areas in YB experienced precipitation, with localized heavy rainstorms. From the above two cases, we can see that the RH in the middle troposphere can be used to distinguish between hail and heavy precipitation during severe convective events.

## 5. Conclusion

This study presents relative humidity (RH) measurements with a developed synergetic algorithm with the combination of Raman lidar, MWR, and satellite at three sites (northern China, middle of China and southern China) from 1 July to 31 November. The methodology for obtaining RH from the synergetic algorithm was introduced. The five-month field campaign was performed and linear regression between the lidar, MWR, satellite, synergetic algorithm and radiosonde data at the range 0-10000 m was presented to testify the accuracy.

Strong correlations of RH values over 0.9 were observed between radiosonde measurements and profiles derived from the synergetic approach at three representative sites in China. The lowest MB values (4.93% in HHHT, 2.63% in YB and 2.40% in QY)are observed when lidar data is integrated into the synergetic algorithm, which highlights the accuracy of the lidar data below 3000 m. However, the MB from lidar increased drastically above this height, which suggests the greater applicability of satellite or MWR in the middle and higher layers. In terms of the seasonal characteristic, QY exhibits the predominant seasonal feature throughout most heights, with peak mean values of 80.65% in July at 1000-2000 m and minimal values of 20.50% in November at 7000-10000 m.

These results validate the capabilities of the newly developed method to deliver accurate measurements of RH information throughout the troposphere. It also explores the potential of satellite data integration for RH profile retrieval for the first time. However, there are still problems with individual data at certain times during the fusing process. For example, there are few effective data filtered by quality control methods for FY4B data. Therefore, the matching accuracy and more high-quality FY4B data will be improved in future development.

**Declaration of Competing Interest**

The authors declare that they have no known competing financial interests or personal relationships that could have appeared to influence the work reported in this paper.

**Data availability**

Raman lidar, MWR, satellite, radiosonde and other auxiliary data used to generate the results of this paper are available from the authors upon request (email: zychen@btbu.edu.cn).

**Acknowledgments**

This work was supported by the Innovation and Development Special Project of China Meteorological Administration (No. CXFZ2024J011 and CXFZ2024J057), National Key Research and Development Program of China (No. 2024YFC3711701) and the project (Simulation of cloud lidar echo signal and study on cloud microphysics characteristics) from Aerospace Information Innovation Research Institute, Chinese Academy of Sciences. The authors thank the colleagues who participated in the operation of the lidar system at our site. We also acknowledge the CMA for the satellite (FY4B) data, radiosonde data (https://ladsweb.modaps.eosd is.nasa.gov), and the European Center for Medium‑Range Weather Forecasts (ECMWF) for the ERA5 reanalysis data (https://climate.copernicus.eu /climate‑reanalysis).

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

**List of Tables**

**Table 1** Instruments and monitoring parameters

| Instrument | Parameters/units | Temporal-spatial Resolution |
|---|---|---|
| Raman lidar | Relative humidity (RH) | 7.5 m, 3 minutes |
| Microwave radiometer (MWR) | Temperature ($^O$C), Relative humidity (RH) | 50 m, 3 minutes |
| FY4B | Relative humidity (RH) | 1 hour |

**Table 2** Assessment of the accuracy of four RH retrieval results (lidar, MWR, satellite and synergetic algorithm) compared with radiosonde at 47 sites in China.

| Comparison with radiosonde | Number of sample | R | MB (%) | MAB (%) | RMSE (%) |
|---|---|---|---|---|---|
| lidar | 192111 | 0.91 | 0.56 | 6.7 | 10.67 |
| MWR | 192111 | 0.82 | -1.49 | 10.79 | 14.31 |
| satellite | 192111 | 0.74 | 1.08 | 13.19 | 17.02 |
| syngenetic algorithm | 192111 | 0.98 | 0.42 | 3.24 | 5.27 |

**Table 3** The same as Table 2 but at three representative sites in China.

| HHHT (northern China) | Comparison with radiosonde | Number of sample | R | RMSE (%) |
|---|---|---|---|---|
| | lidar | 3771 | 0.83 | 20 |
| | MWR | 3771 | 0.74 | 25 |
| | satellite | 3771 | 0.76 | 24 |
| | syngenetic algorithm | 3771 | 0.94 | 10 |
| YB (middle China) | lidar | 7542 | 0.86 | 19 |
| | MWR | 7542 | 0.80 | 26 |
| | satellite | 7542 | 0.83 | 29 |
| | synergetic algorithm | 7542 | 0.92 | 12 |

| QY | lidar | 8110 | 0.86 | 18 |
| (southern China) | MWR | 8110 | 0.75 | 19 |
| | satellite | 8110 | 0.66 | 21 |
| | synergetic algorithm | 8110 | 0.93 | 11 |

**List of figures**

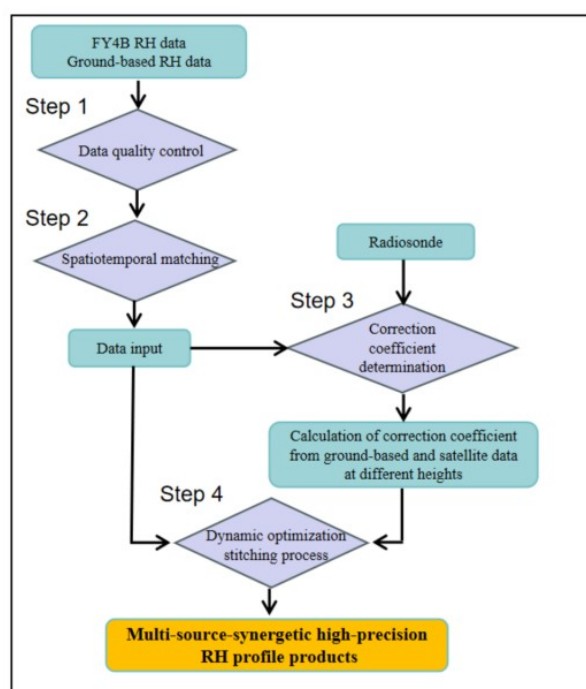

**Fig. 1** Sketch of the retrieval scheme. Details are given in the text.

**Correction coefficient dynamically adjusted according to the latest time data**

**Real-time observation data**

Radiosonde data $R_{radio}$
Lidar data $R_{Lidar}$
MVR data $R_{MWR}$
Satellite FY4B data $R_{satellite}$

Lidar data deviation $D_{Lidar}=R_{Lidar}-R_{radio}$
Lidar data correction coefficient $C_{Lidar}$
$C_{Lidar}=(|D_{satellite}|+|D_{MWR}|)/[2*(|D_{satellite}|+|D_{MWR}|+|D_{Lidar}|)]$

MVR data deviation $D_{MWR}=R_{MWR}-R_{radio}$
MVR data correction coefficient $C_{MWR}$
$C_{MWR}=(|D_{satellite}|+|D_{Lidar}|)/[2*(|D_{satellite}|+|(D_{MWR}|+|D_{Lidar}|)]$

FY4B data deviation $D_{satellite}=R_{satellite}-R_{radio}$
FY4B data correction coefficient $C_{satellite}$
$C_{satellite}=(|D_{MWR}|+|D_{Lidar}|)/[2*(|D_{satellite}|+|D_{MWR}|+|D_{Lidar}|)]$

**Synergy of ground-based remote sensing and satellite data**

Synergetic data
$S_{RH}=R_{satellite}*C_{satellite}+R_{MWR}*C_{MWR}+R_{Lidar}*C_{Lidar}$

**Fig. 2** The dynamic optimal stitching process

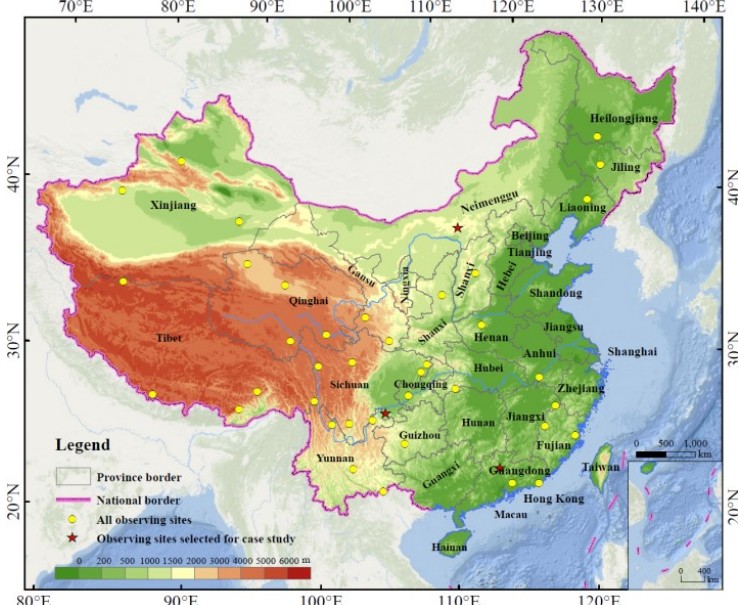

**Fig. 3** The observing sites (yellow circles) and three selected sites (red stars) for statistics and

case studies are marked in the

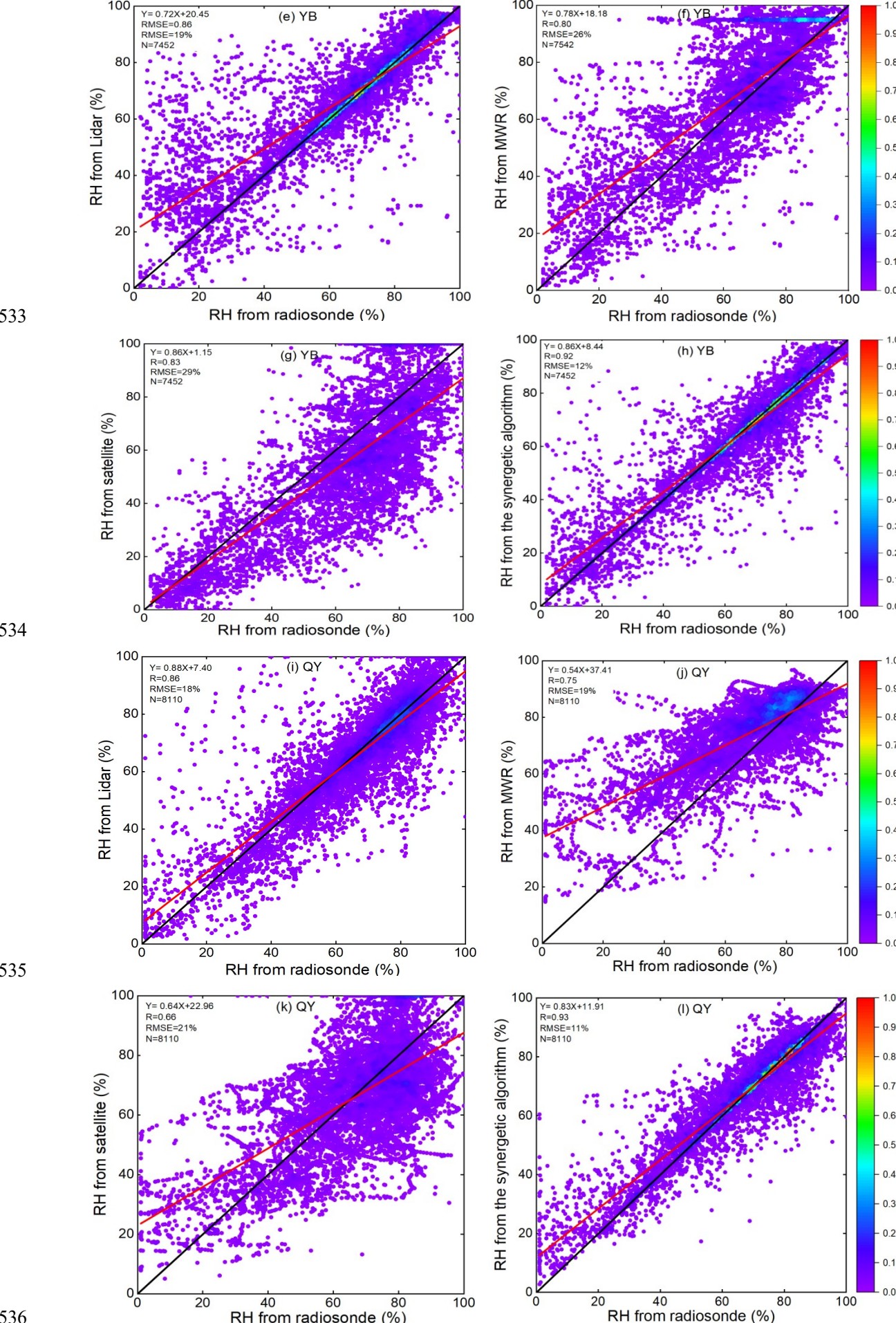

**Fig. 4** Four-methods-retrieved RH results (lidar, MWR, satellite and synergetic algorithm)

compared with radiosonde at three sites in China from 1 July to 31 November 2024. (a)

Comparison between lidar and radiosonde in HHHT, (b) Comparison between MWR and

radiosonde in HHHT, (c) Comparison between satellite and radiosonde HHHT, (d)

Comparison between synergetic algorithm and radiosonde in HHHT; (e)-(h), the same as (a)-(d)

but in YB. (i)-(l), the same as (a)-(d) but in QY. The red line shows the regression line. The

black line is the one-to-one line.

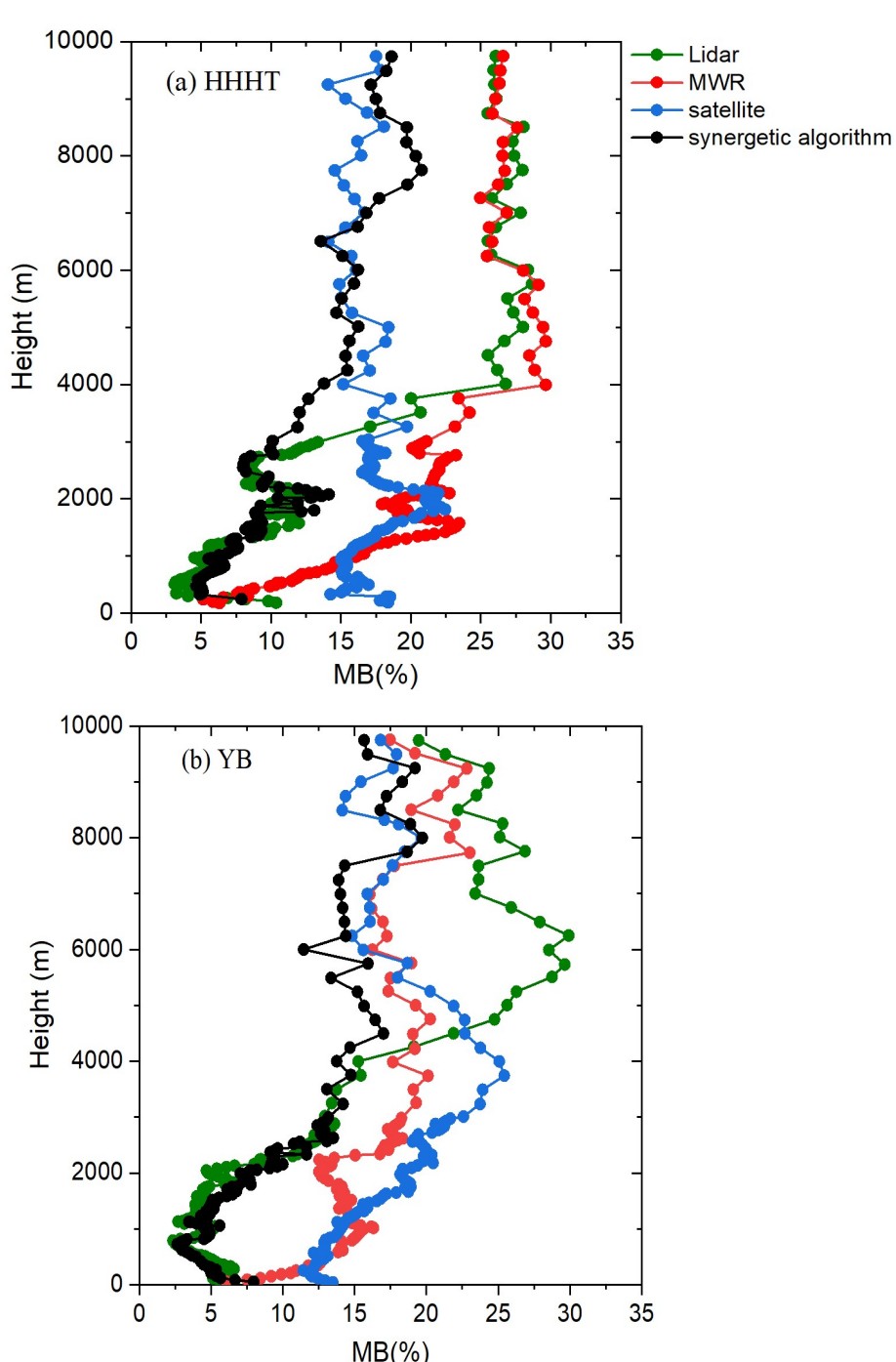

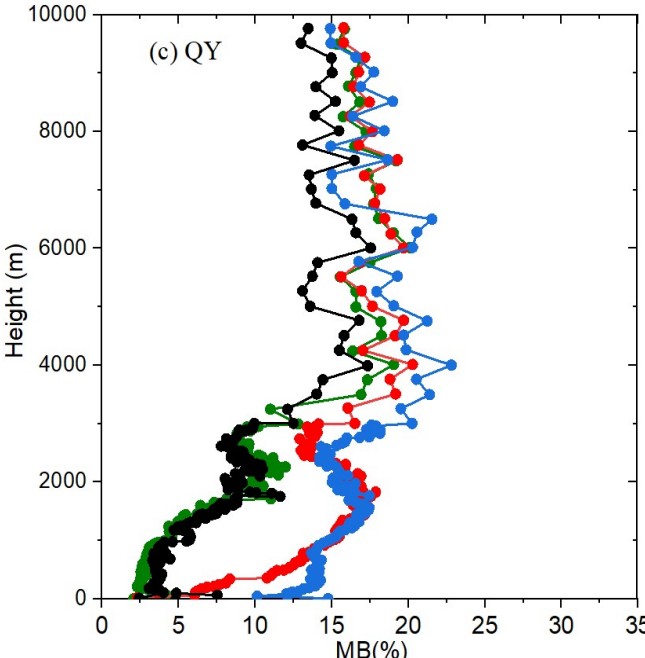

**Fig. 5** RH vertical mean bias (MB) profiles retrieved from lidar, MWR, satellite and synergetic

algorithm compared to the radiosonde data in (a) HHHT, (b) YB and (c) QY.

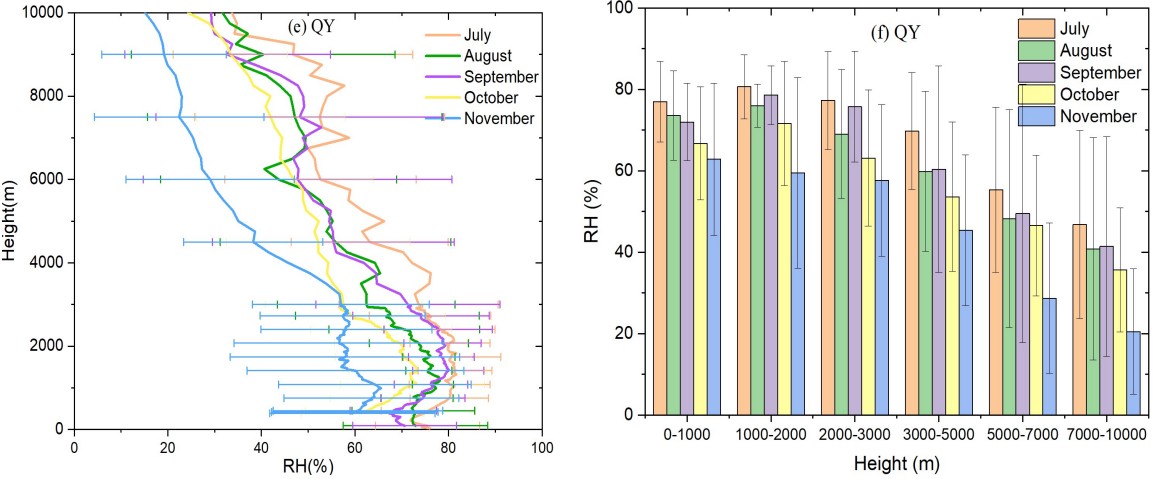

**Fig. 6** RH Monthly vertical profiles (left) and monthly mean values for different heights (right)

in (a)-(b) HHHT, (c)-(d)YB and (e)-(f) QY. The error bars indicate the standard deviation.

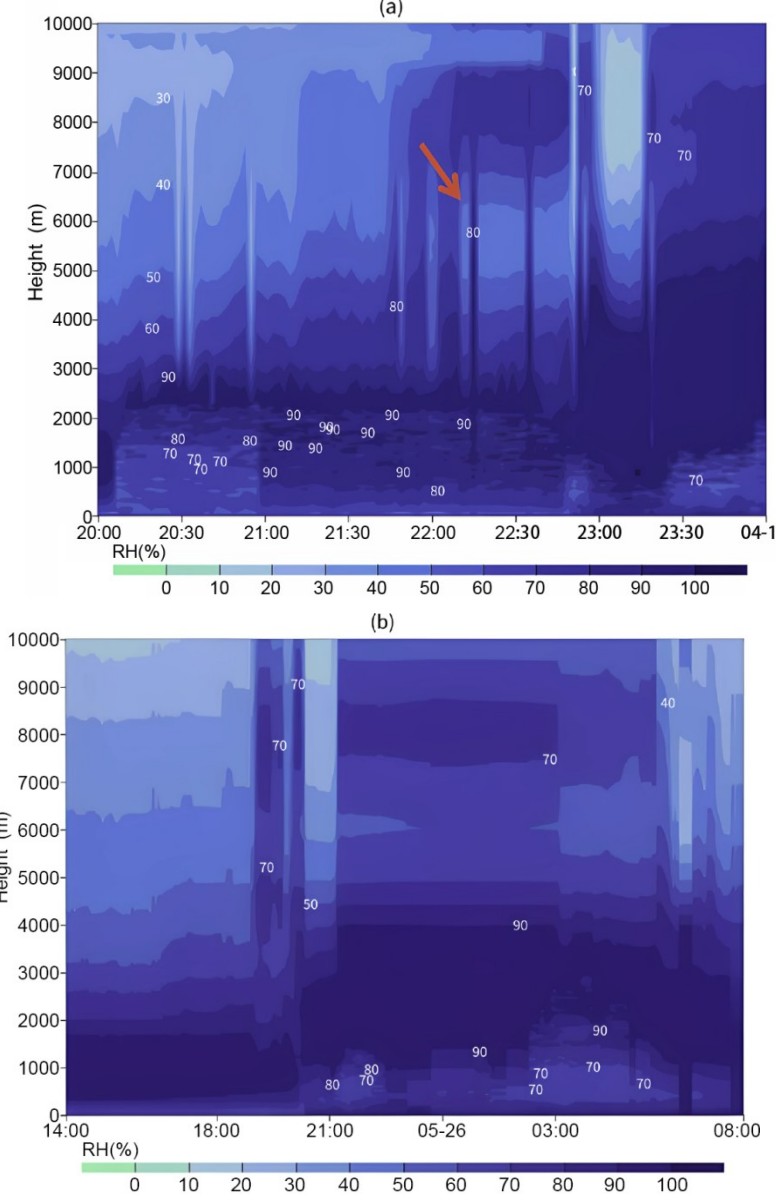

**Fig. 7** Height-time display of RH from the synergetic retrieval during two convective cases (a)

from 20:00 LST to 23:30 LST 15 April and (b) from 14:00 LST 25 May to 08:00 LST 26 May

2024 in YB. The red arrow indicates the less humidity in the layer when the hailfal occurred in

the first convective case.