# Peer review of "A new method to retrieve relative humidity profiles from a synergy of Raman LiDAR, microwave radiometer and satellite"

_EGUsphere, 2025_

## Referee Comment (RC2)

**General comments to authors:**

The manuscript presents a new method for obtaining continuous relative humidity distribution by integrating multiple data. And it was validated through observation data from 47 stations in China over a period of five months. The results showed good consistency between the use of synergetic algorithm and radiosonde data. The additional monthly statistical analysis and case studies have also expanded the practical application scenarios of this method. I think this manuscript can be published in the journal Atmospheric Measurement Technology. But before that, I think it is necessary to answer the following questions and make minor modifications. But before that, I think it is necessary to answer the following questions and make minor modifications.

**Specific comments:**

1.  In the introduction section, the author discusses the previous methods of using multi-source data such as radar to study relative humidity profiles, but does not explain how these differ from the synergetic algorithm mentioned in the manuscript. Please explain specifically where the proposed methods are new? What is the difference from before?

2.  The author mentioned in the introduction section (lines 69-76) that many literature studies have introduced data from Raman LiDAR and microwave radiometer to obtain continuous RH profile data, but did not elaborate on the differences between your method and theirs. This will confuse readers: where is your new method new? Please provide a supplementary description for this section.

3.  In section 2.1, when it comes to Raman differential absorption and setting the signal-to-noise ratio to 3, the description is unclear.

4.  In the Instrumentation section, line 118 mentions' The uncertainty of the instrument can reach a confidence level of 95.5%. '. This description is confusing.

5.  In the Methods and Evaluation section, the core steps of the dynamic optimal stitching algorithm (Figure 2) mentioned, such as correction coefficient calculation and weight allocation, lack mathematical formulas or quantitative descriptions. Suggest adding specific algorithm formulas and detailed explanations in the text rather than in the figure.

6.  There is a formatting issue with Table 2, please make the necessary changes.

7.  I noticed that after introducing the observation results of LiDAR, the maximum correlation coefficients R of the collaborative algorithm in HHHT, YB, and QY were 0.90, 0.91, and 0.93, respectively, which is very good. However, the RMSE of each instrument's individual data and sounding data exceeds 20% (Table 3). Does this indicate poor reliability of the data? What do you think about this?

8.  At the end of section 4.1, the author analyzed the sources of errors. There are many sources of error analysis that lead to data uncertainty, such as the consistency of observation equipment? What is the uncertainty caused by regional differences? What is the physical essence that leads to differences here? There are many things worth pondering, which is why I recommend publishing this manuscript. It is meaningless to simply analyze these data differences.

9.  The case analysis relies on ERA5 reanalysis data to provide the weather circulation situation (Figure 7), and the results generated by the synergetic method with Figure 8 lack correlation explanation. This is very confusing.

10. The conclusion section is cumbersome and not concise, please rephrase.

---

## Author Comment (AC1)

**Response to reviewers**

This study evaluates relative humidity (RH) profiles at 47 stations in China using a synergetic multi-source algorithm (dynamic optimal stitching algorithm), comparing the results with radiosonde data. The results suggest that the synergetic algorithm outperforms individual instruments (lidar, microwave radiometer, and satellite) at various altitudes, particularly by leveraging lidar data in the lower atmosphere (<3000 m) and satellite data in the upper atmosphere (>3000 m). The study is valuable and aligns with the scope of the journal. But some concerns should be addressed before acceptance:

1. lines 57 and 60, the definition of MVR/MWR is conflict? Please make it clear, also in section 2.2.
Reply: We have corrected it as "MWR" through the paper.

2. The introduction could be enhanced. Although the introduction discusses the integration of Raman lidar and MWR for simultaneous RH profile retrieval, with multiple references cited, the authors do not clearly distinguish how their proposed method differs from existing techniques. The novelty should be emphasized.

Reply: Yes, some work has focused on the integration of Raman lidar and MWR for RH retrieval. However, most of their algorithms primarily utilize statistical methods, performing data fusion between different instruments based on long-term time-series data from individual locations. While these approaches are suitable for observations at single stations, they lack universality when applied to scenarios requiring data integration from multiple sites or broader geographical coverage. Moreover, replacing instruments or equipment may also introduce additional inconsistencies. Compared to these existing techniques, our new method not only incorporates satellite data but also dynamically determines optimal fusion coefficients, enabling device model independence and geographical adaptability. Thus it eliminates constraints imposed by equipment specifications or observation locations, ensuring broad applicability across diverse scenarios.

We have clarified it in the revised paper.

3. Section 3.1 requires more detailed explanation of the proposed method.

Specifically, the formula for calculating the correction coefficients should be included in the main text, not just in Figure 2. Furthermore, using radiosonde as a reference and applying deviations from other measurements as weighting coefficients—how does this differ from traditional data assimilation methods, and what are the advantages?

Reply: Thank you for the comments. We have included the calculation of the correction coefficients in the main text. Compared to the traditional data assimilation methods, the weighting coefficients are dynamically determined by comparing the deviations from other measurements with the reference of radiosonde, and it can guarantee the independence of each device and observing site. So the new algorithm is real-time calibrated. Besides that, it incorporates the satellite data which is more trustworthy in the higher layer of RH retrieval. Thus it can ensure broader applicability with higher precision.

We have clarified it in the revised paper.

4. Are the biases between other measurements and radiosonde data time-dependent? How do these biases differ from the theoretical measurement errors of the instruments? This can be further discussed.

Reply: Yes, the bias between other measurements and radiosonde data is time-dependent. In addition to the theoretical measurement errors of the instrument, the errors also stem from the other sources. First, although all instruments are co-located in the ground, radiosondes deviate at higher heights, and uncertainty increases if clouds are present. Second, satellites provide gridded data, requiring the selection of ground observation points closest to its grid's latitude and longitude, which introduces uncertainty. Finally, errors during the retrieval process (e.g., neural networks for MWR) are also unavoidable.

We have clarified it in the revised paper.

5. The results from the synergetic algorithm in Figure 5 appear to outperform the best observational data. How can this be explained in terms of the algorithm formula presented in Figure 2?

Reply: Yes, because each single instrument has its limitations. For example, Lidar has

low signal-to-noise ratio and is greatly affected by clouds at higher altitudes, while MWR and satellites are passive remote sensing, resulting in high uncertainty in inversion results. So the core of the new-developed synergetic algorithm is to find the optical fusion combing the best performance of each instrument at different heights. For example, the lidar data is incorporated into the synergetic algorithm at lower heights. Thus the synergetic method retrieved RH could provide more accurate results compared to the other single source data. And its performance is superior to the best observational data as conventionally used in the experiment.

6. Some statements are unclear and confusing. For example: "But the signal-to-noise ratio (SNR) decreases with height, thus the threshold of SNR should be set." The purpose of setting an SNR threshold is to ensure signal reliability, not simply because SNR decreases with height. Furthermore, why is the threshold set to 3? Please provide references to support this choice.

Reply: Thank you for the comments. Based on our extensive comparisons with radiosonde data, empirical evidence from CMA's long-term observations indicates that selecting lidar signals with the signal-to-noise ratios (SNR) >3 can significantly improve the consistency between retrieved RH profiles and radiosonde measurements.

We have clarified it in the revised paper.

7. The font size in the figures is too small and should be adjusted for better readability.

Reply: We have redrawn the figures.

---

## Author Comment (AC2)

**Response to reviewers**

General comments to authors:

The authors introduced a new method in the manuscript to obtain continuous relative humidity profiles by integrating data from Raman LiDAR, microwave radiometer, and satellite sources. And it was validated through five months of observation data from north to south in China, which showed a high consistency with radiosonde data. Monthly statistical analysis and case studies also demonstrated the applicability of this method. I think this manuscript can be published in the journal Atmospheric Measurement Techniques. But before that, I think it is necessary to answer the following questions and make minor modifications.

Specific comments:

1. In the introduction section: the authors mentioned that the observation data from China's Fengyun (FY) satellite is rarely used for comprehensive inversion of relative humidity information, but did not mention the use of satellite data from other countries for inversion of relative humidity information. Please supplement this section.

Reply: Thank you for the comments. We have included more references to the use of satellite data from other countries for relative humidity information inversion in the Introduction part.

2. The author mentioned in the introduction section (lines 69-76) that many literature studies have introduced data from Raman lidar and microwave radiometer to obtain continuous RH profile data, but did not elaborate on the differences between your method and theirs. This will confuse readers: where is your new method new? Please provide a supplementary description for this section.

Reply: Yes, some work has focused on the integration of Raman lidar and MWR for RH retrieval. However, most of their algorithms primarily utilize statistical methods, performing data fusion between different instruments based on long-term time-series data from individual locations. While these approaches are suitable for observations at single stations, they lack universality when applied to scenarios requiring data integration from multiple sites or broader geographical coverage. Moreover, replacing instruments or equipment may also introduce additional inconsistencies. Compared to these existing techniques, our new method not only incorporates satellite data but also dynamically determines optimal fusion coefficients, enabling device model independence and geographical adaptability. Thus it eliminates constraints imposed by equipment specifications or observation locations, ensuring broad applicability across diverse scenarios.

We have provided a more detailed description for this part (in section 1 and section 3).

3. In section 2.1, when it comes to Raman differential absorption and setting the signal-to-noise , the description is unclear.

Reply: The threshold value of the signal-to-noise ratio is set as 3 based on our extensive comparisons with radiosonde data from CMA's long-term observations. The results indicate that selecting lidar signals with signal-to-noise ratios (SNR) >3 can significantly improve the consistency between retrieved RH profiles and radiosonde measurements. We have clarified it in this section.

4. In the Instrumentation section, line 118 mentions' The uncertainty of the instrument can reach a confidence level of 95.5%. '. This description is confusing.

Reply: We have deleted the sentence.

5. In the Methods and Evaluation section, the core steps of the dynamic optimal stitching algorithm (Figure 2) mentioned, such as correction coefficient calculation and weight allocation, lack mathematical formulas or quantitative descriptions. Suggest adding specific algorithm formulas and detailed explanations in the text rather than in the figure.

Reply: We have included the calculation of the correction coefficients by adding the mathematical formulas and descriptions in this part. Because the weighting coefficients are dynamically determined by comparing the deviations from other measurements with the reference of radiosonde, it can guarantee the independence of each device and observing site. It highlights that the new algorithm is real-time calibrated.

6. There is a formatting issue with Table 2, please make the necessary changes.

Reply: We have corrected it.

7. I noticed that after introducing the observation results of LiDAR, the maximum correlation coefficients R of the collaborative algorithm in HHHT, YB, and QY were 0.90, 0.91, and 0.93, respectively, which is very good. However, the RMSE of each instrument's individual data and sounding data exceeds 20% (Table 3). Does this indicate poor reliability of the data? What do you think about this?

Reply: We have checked the data and found that the precipitation data has not been removed from our dataset. This has led to the relatively larger RMSE. We have removed the precipitation data and recalculated the data.

8. At the end of section 4.1, the author analyzed the sources of errors. There are many sources of error analysis that lead to data uncertainty, such as the consistency of observation equipment? What is the uncertainty caused by regional differences? What is the physical essence that leads to differences here? There are many things worth pondering, which is why I recommend publishing this manuscript. It is meaningless to simply analyze these data differences.

Reply: Thank you for the comments. For the error sources, in addition to the theoretical measurement errors of the instrument, the errors also stem from the other sources. First, although all instruments are co-located in the ground, radiosondes deviate at higher heights, and uncertainty increases if clouds are present. Second, satellites provide gridded data, requiring the selection of ground observation points closest to its grid's latitude and longitude, which introduces

uncertainty. Finally, errors during the retrieval process (e.g., neural networks for MWR) are also unavoidable.

The results in three observing sites (HHHT, YB and QY) show a similar RMSE with a value of 10% -12 %. It indicates the relatively good regional universality of the synegetic algorithm. Though there is no obvious uncertainty caused by regional differences, we found that QY exhibits the predominant seasonal feature throughout most heights. In contrast, no discernible seasonal characteristics in RH profiles are observed in HHHT or YB. Thus we believe diverse atmospheric circulation patterns and geographical environments could result in regional variations in RH values.

We have clarified it in the section 4.2.

9. The case analysis relies on ERA5 reanalysis data to provide the weather circulation situation (Figure 7), and the results generated by the synergetic method with Figure 8 lack correlation explanation. This is very confusing.

Reply: Thank you for the comments. We have chosen new cases and rewritten the case analysis. In the revised version, we analyzed the RH evolution in two severe convection cases. By comparing the characteristics of the height-time RH, we find that the RH in the middle troposphere is critical for distinguishing between hail and heavy precipitation. These two new cases can better demonstrate the importance of obtaining accurate spatio-temporal RH information for weather prediction.

10. The conclusion section is cumbersome and not concise, please rephrase.

Reply: We have shortened the conclusion part to make it more concise.

---

## Referee Report (RR1)

**General comments to authors:**

The authors proposed a synergistic algorithm based on five months of observation data in China. And through application research and case analysis on different sites, we have seen the application prospects of this method, which I think is a very interesting study. However, before publication, I feel that there are still some areas that can be slightly revised and improved.

**Specific comments:**

1. Throughout the manuscript, 'lidar' should be 'LiDAR'.
2. The language expression of the authors needs minor revisions, especially in the introduction section where there are too many long sentences, which can confuse and obscure readers.
3. Some abbreviations that first appear in the manuscript require brief explanations, such as' SAPHIR ',' MEGHA-TROPIQUES', ' LST', etc.
4. Why are the data samples from three different sites in different regions of China not consistent in Table 3 (3771,7542,8110, respectively). Is there any special significance to this? If not, is the sample data consistent and comparable? If not, is the result convincing?
5. To enhance visual harmony, we recommend unifying the dimensions of the subfigures in Figures 6 and 7.
6. The authors re-selected the case analysis, but it was not reflected in the final conclusion section. Additionally, the abstract is not concise enough, and I believe it is necessary to rephrase it.

---

## Author Response (AR2)

**Response to Reviewer**

General comments to authors:

The authors proposed a synergistic algorithm based on five months of observation data in China. And through application research and case analysis on different sites, we have seen the application prospects of this method, which I think is a very interesting study. However, before publication, I feel that there are still some areas that can be slightly revised and improved.

Specific comments:

1. Throughout the manuscript, 'lidar' should be 'LiDAR'.

Reply: We have corrected it throughout the paper.

2. The language expression of the authors needs minor revisions, especially in the introduction section where there are too many long sentences, which can confuse and obscure readers.

Reply: We have rephrased the language expression of the paper, especially the introduction section.

3. Some abbreviations that first appear in the manuscript require brief explanations, such as' SAPHIR ',' MEGHA-TROPIQUES', ' LST', etc.

Reply: We have added the explanations of the abbreviations through the paper.

4. Why are the data samples from three different sites in different regions of China not consistent in Table 3 (3771,7542,8110, respectively). Is there any special significance to this? If not, is the sample data consistent and comparable? If not, is the result convincing?

Reply: The data samples are different because of two reasons. Firstly, the data samples from the instruments (LiDAR, MWR, satellite) at three sites are different through quality control. Secondly, after removing precipitation data, the sample numbers also vary. We believe that when the sample size reaches a certain level, such

as thousands, the comparison has a certain representative significance. We have clarified it.

5. To enhance visual harmony, we recommend unifying the dimensions of the subfigures in Figures 6 and 7.

Reply: We have unified it.

6. The authors re-selected the case analysis, but it was not reflected in the final conclusion section. Additionally, the abstract is not concise enough, and I believe it is necessary to rephrase it.

Reply: We have added the explanation of the case result in the conclusion part. We also rephrased the abstract.